# Inflammation and Aging of Hematopoietic Stem Cells in Their Niche

**DOI:** 10.3390/cells10081849

**Published:** 2021-07-21

**Authors:** Daozheng Yang, Gerald de Haan

**Affiliations:** 1European Research Institute for the Biology of Ageing (ERIBA), University Medical Center Groningen, University of Groningen, 9713 AV Groningen, The Netherlands; d.yang@umcg.nl; 2Sanquin Research, Landsteiner Laboratory, Amsterdam UMC, 1006 AD Amsterdam, The Netherlands

**Keywords:** hematopoietic stem cells, aging, niche, heterogeneity, inflammation

## Abstract

Hematopoietic stem cells (HSCs) sustain the lifelong production of all blood cell lineages. The functioning of aged HSCs is impaired, including a declined repopulation capacity and myeloid and platelet-restricted differentiation. Both cell-intrinsic and microenvironmental extrinsic factors contribute to HSC aging. Recent studies highlight the emerging role of inflammation in contributing to HSC aging. In this review, we summarize the recent finding of age-associated changes of HSCs and the bone marrow niche in which they lodge, and discuss how inflammation may drive HSC aging.

## 1. Age-Associated Changes in HSCs

### 1.1. The Functioning of Hematopoietic Stem Cells Declines with Age

Multiple aging-associated hematopoietic features occur in both peripheral blood (PB) and bone marrow (BM). Aged mice display elevated platelet counts [1,2,3] and myeloid cells [4], with concomitant decreased red blood cells (RBC) [1,3] and haemoglobin (Hb) [1,3]. In aged BM, there is an increase in the frequency of committed megakaryocyte progenitors (MkPs) [1,2,5,6] and granulocyte-monocyte progenitor (GMP) [2,6] and a decline in the frequency of common lymphoid progenitor (CLP) [6,7] and colony forming unit-erythroid (CFU-E) [2,6]. Elderly humans frequently suffer from anemia, with a ~25% prevalence of anemia found in people over 85 years [8]. It seems plausible that the alterations in hematopoiesis during aging are largely due to changes that first occur in the cells from which all blood cells derive, the hematopoietic stem cells.

HSCs can undergo both asymmetric and symmetric cell division [9,10] and have the ability to differentiate into multiple mature blood cells and to self-renew, thus sustaining the stem cell pool throughout life. In general, most HSCs are quiescent and divide very rarely (~4 times) throughout adult life [11]. Whereas young HSCs primarily divide asymmetrically, aged HSCs undergo mainly symmetric divisions [12]. With aging, HSCs display functional decline (Figure 1). Aged HSCs show skewed myeloid differentiation potentials at the expense of the reduced production of lymphocytes [1,13,14,15,16,17]. In addition, serial transplantation studies have convincingly demonstrated that aged HSCs display reduced engraftment and self-renewal [1,16,18]. In mice, the HSC pool size expands with age, which is considered as a compensatory effect to adjust for the loss of cellular potential.

### 1.2. HSCs Become More Heterogeneous upon Aging

It is now well established that clonal hematopoiesis is a frequent event in otherwise healthy elderly individuals [19]. This indicates that, during aging, the clonal composition of the human HSC pool becomes altered and increasingly heterogeneous. In murine models, a series of studies have assessed the clonal composition of HSCs. These were done by either barcoding [20,21,22] or single cell transplantation [1,16,23]. HSC barcoding can be achieved by virally inserting a random DNA sequence [20,22] or the excision and inversion of endogenous Cre-mediated DNA elements [24]. Combined with multiplex high-throughput sequencing, the clonal behavior of HSC can be assessed quantitatively. We previously reported that the young HSC pool is dominated by large clones, while aged HSC pool is composed of more, but smaller clones [20]. Single HSC transplantations have identified clones with restricted lineage potential. Beyond now classical myeloid- or lymphoid-biased HSCs [20], HSCs with additional lineage restriction patterns have also been identified [1,23,25]. Interestingly, in aged bone marrow, a “latent” HSC population was identified that showed a myeloid uni-lineage restriction in primary transplantation but multipotent differentiation in secondary transplantation [1]. The issue thus arises whether HSCs with distinct repopulation patterns can be prospectively isolated, and if so, which cell surface markers uniquely identify HSC subpopulations with heterogeneous function.

Multiple sorting strategies have been developed to purify primitive HSCs from mouse bone marrow [26,27,28]. These sorting strategies all allow for the isolation of highly purified HSCs that, at the single cell level, can sustain long term self-renewal and differentiation upon transplantation. However, even when highly purified, and irrespective of marker expression, not all HSCs are identical. Multiple markers are heterogeneously expressed on highly purified HSCs and numerous studies have investigated functional differences between HSC subpopulations with such heterogeneous expression of different markers (Table 1). Lymphoid-biased HSCs, displaying high self-renewal capacity, are usually marked by CD86 [29], Jam2 [30] and Satb1 [31,32,33], while myeloid/platelet-biased HSCs (marked by CD41 [34,35,36,37,38], Neo1 [39,40], CD61 [41], IL27Ra [42], Alcam [43,44] and c-Kit [45]) possess low self-renewal capacity. Of note, although CD150^high^ [46] or vWF^+^ [5,47] HSCs display high self-renewal, they show myeloid/platelet bias and are able to generate other lineage-biased HSCs, indicating that these cells are on the top of the HSC hierarchy.

Interestingly, many markers that are used to purify HSCs are either upregulated or downregulated upon HSC aging. Alterations in the expression of these markers correlate with functional activity (Table 1). For example, CD150, one of the SLAM markers that is frequently used to sort HSCs, predominantly marks myeloid-biased HSCs and is preferentially expressed on aged HSCs [41,46]. Similarly, CD86, a marker of lymphoid-biased HSCs, is downregulated upon aging [29]. The upregulation of CD150 and the concomitant downregulation of CD86 is consistent with the myeloid-biased phenotype of aged HSCs. In other words, an expansion of CD150^high^ HSCs and a reduction in CD86^high^ HSCs can explain the functional decline of aged HSCs and also reflect changes in the heterogeneity of aged HSCs. Importantly, this also suggests that although the aged HSCs pool is dominated by myeloid-biased HSCs, balanced or “young-like” HSCs that contain high stem cell potential are still present, which could potentially be identified using aging-associated markers. For example, we would predict that aged CD150^low^ and CD86^high^ HSCs behave similar young-like HSCs. It is likely that additional markers are co-expressed by myeloid-biased HSCs, while others are co-expressed by lymphoid-biased HSCs. However, although several single-HSC RNA-sequencings have been performed, the full transcriptome of myeloid- versus lymphoid-biased HSCs remains to be determined.

At present, the molecular origin of HSC heterogeneity remains obscure [53]. However, it is conceivable that distinct HSCs localize to distinct niches so that diverse cell-extrinsic signals may be delivered to individual HSCs [53]. Below, we discuss the regulation of the HSC niche and how the niche may alter with age.

## 2. Age-Associated Changes in the Bone Marrow Niche

HSCs are located in the BM and are surrounded by multiple cell types. These surrounding cells, together with the extracellular matrix, form a complex microenvironment, known as the HSC niche. The niche produces diverse cytokines, extracellular matrix proteins and adhesion molecules that regulate HSC survival, proliferation, self-renewal and differentiation [54]. Recently, it was found that N-cadherin-expressing bone and marrow stromal progenitor cells remain dormant HSCs in homeostasis and post-chemotherapy [55]. In addition, megakaryocytes were found to be associated with platelet and myeloid-biased vWF^+^ HSCs, while vWF^-^ lymphoid-biased HSCs are regulated by nestin and neural–glial antigen 2^+^ (NG2^+^) arteriolar niche cells [56]. These findings illustrate how the functional heterogeneity of HSCs may result from distinct locations in the niche in the steady-state hematopoiesis.

However, to what extent the aging of the niche causally contributes to HSC aging is unclear. Although age-associated phenotypes cannot be rescued when aged HSCs are transplanted into lethally irradiated young mice, young HSCs display impaired engraftment and reduced T cell production when transplanted into old recipients compared to those transplanted into young recipients [57,58]. This supports the hypothesis that an aged BM niche contributes to reduced self-renewal and skewed myeloid differentiation of aged HSCs. The impact of an aged niche on HSC functionality has been assessed by transplantations, where recipients were usually lethally irradiated [57,58]. Lethal irradiation creates space for donor HSCs, but affects, and possibly destroys, the BM niche [59,60]. Thus, this experimental setup ignores putative effects of different radiosensitivity of young and aged BM niches. Therefore, irradiation-based preconditioning makes it difficult to evaluate the direct contribution of age-related HSC niche changes to HSC functions [61]. To circumvent this problem, an excess of aged HSCs has been transplanted into young recipients, without preconditioning. Interestingly, no significant functional rejuvenation of aged HSCs was seen, but the transcriptome profile of aged HSCs was restored [61]. Although this suggests that aged HSCs are not rejuvenated by a young niche, it does imply that age-associated extrinsic changes are relevant for HSC aging. To address this, young HSCs would have to be transplanted into non-conditioned aged BM, to assess whether young HSCs are functionally aged.

It is of great relevance to understand how the aging of the niche may be involved in HSC aging. A growing body of evidence suggests that the BM niche does change in aged mice (Figure 1) [4,62,63]. For example, it has been shown that HSCs are re-localized in the aging niche, where aged HSCs are away from arterioles or megakaryocytes but close to perivascular Nestin-GFP^dim^ cells and sinusoids compared to young counterparts [62,64]. While it has been suggested that the overall vasculature volume and the endothelial area occupancy is not altered in aged BM, other studies found that aging imposed drastic remodeling of BM vascular architecture, as evidenced by an overall increase in vascular density. Interestingly, during aging, different vascular segments undergo different changes [4,62,63]. Endosteal vessels, transition zone vessels and α-smooth muscle actin-positive (α-SMA^+^) arteries were reduced in aged BM, whereas sinusoidal areas appeared unchanged [4,62,63]. Consistent with the decrease of arteries, type H capillaries and stem cell factor (SCF) levels were also reduced [63]. In contrast, the CD31^high^ endomucin (EMCN)^−^ capillaries located further from bone increased with age [4].

In addition to changes of the vascular structure, BM vascular endothelial cells (ECs) also alter with age. The overall number of CD45^−^Ter119^−^CD31^high^ ECs are increased whereas CD45^−^Ter119^−^CD31^high^Sca-1^high^ arteriolar ECs are decreased with age [62]. Aged ECs display various dysfunctions, which are associated by increased vascular leakiness, increased reactive oxygen species (ROS) and decreased in vitro angiogenic potential, suggesting that the instructive endothelial niche function is compromised during aging [65].

BM mesenchymal stromal cells (MSCs) are crucial elements involved in the maintenance of hematopoiesis. The depletion of MSCs impaired HSC homing and expansion [66,67]. Although overall the number of MSCs is expanded with age [3,62], MSCs derived from old mice exhibited lower clonogenic potential in vitro [58,62]. NG2^+^ perivascular cells are reduced in aged BM, which explains the reduction of vWF^-^ lymphoid-biased HSCs with age [56,62,63]. The expression of important niche factors such as *Cxcl12*, *Scf*, and *Angpt1*, was also reduced in aged MSCs [62], indicating a functional decline of aged MSCs. In addition, MSCs in aged mice presented with reduced levels of Osteopontin (OPN), which is associated with the negative regulation of the HSC pool size [68]. Furthermore, the BM niche of OPN knock-out mice mimics aged niches, and the expansion of HSCs and reduced self-renewal were observed when young HSCs were transplanted to OPN deficient recipients [58].

The overall changes of neuronal cells in the BM are contradictory. A reduction in the density of tyrosine hydroxylase^+^ (TH^+^) cells in aged BM has been described [62], while another study suggested an increase [4]. Nevertheless, the denervation of young BM induced premature HSC and niche aging [62]. Sympathetic nervous system (SNS) nerves locally deliver noradrenaline, targeting β2 (ADRβ2) and β3 (ADRβ3) adrenergic receptors that are expressed by both hematopoietic and nonhematopoietic BM cells [62]. ADRβ3 signals are associated with HSC aging, which is evidenced by using of either ADRβ3 agonist or ADRβ3 deficient mice. It has been shown that the treatment of mice with an ADRβ3 agonist increased donor engraftment, and the deletion of ADRβ3 led to the premature aging of HSC [4,62]. 

Megakaryocytes (Mks) regulate HSC quiescence through CXCL4 and TGFβ1 [69,70]. Of note, Mks are associated with myeloid- and platelet-biased vWF^+^ HSCs [56]. With age, the number of Mks increase [64], which is likely responsible for the elevated TGFβ1 and expansion of vWF^+^ HSCs in aged BM [2]. BM macrophages and plasma cells have also been shown to contribute to HSCs aging. Aged BM macrophages expand platelet-biased HSCs while plasma cells stimulate myelopoiesis in vitro [3,71]. In addition, upon aging, both cell types displayed inflammation-associated changes [3,71]. Indeed, a recent study suggests that niche cells display inflammatory transcriptional programs as they age [72]. This observation supports the hypothesis that age-associated inflammation in BM niche may contribute to HSC aging. Below, we will discuss how inflammation may affect HSC functioning during aging.

## 3. Inflammaging of HSC

With age, a chronic, systemic and low grade inflammatory process is referred to as inflammaging, which is associated with immunosenescence and age-related diseases [73,74]. There is evidence indicating that inflammaging occurs in hematopoiesis under chronic inflammatory stress. Interestingly, inflammation-associated stress hematopoiesis is very similar as age-associated hematopoiesis, discussed above. For example, chronic inflammatory signals cause the expansion of HSC and GMP and a reduction of CLP and RBC [36,48]. Similar phenotypes emerge during experimental spondyloarthritis in which mice developed non-resolving inflammation [75]. In addition, consecutive injections of LPS dramatically suppressed erythropoiesis [76]. Most importantly, increasing evidence demonstrates that multiple pro-inflammatory cytokines, including IL-1β [3,4], TNF-α [42], IL-6 [4] and TGFβ1 [2], are present at increased levels in aged BM (Table 2). The inhibition of both IL-1β and TNF-α in aged mice attenuated myelopoiesis [71]. This indicates that aged HSCs are exposed to a niche containing more pro-inflammatory cytokines, which likely contributes to the age-associated HSC dysfunctions. This also suggests that the inhibition of inflammatory responses may rejuvenate aged HSC.

### 3.1. HSCs Are Transiently Activated under Chronic Inflammation

HSCs proliferate in response to both interferons (IFNs) type-I (IFN-α/β) [34,77,78,79] and type-II (IFN-γ) [81], interleukin (IL)-1 [36], tumor necrosis factor (TNF)-α [37], G-CSF [87] and TLR ligands [86] (Table 2). IFN-α, for example, induces cell cycle entry by suppressing the expression of cyclin-dependent kinase inhibitors (CDKIs), which results in decreased expression of *Cdkn1b* (*p27*) and *Cdkn1c* (*p57*) [78,79]. After chronic exposure of HSC to LPS, IL1-β or IFN-γ enhanced proliferation is induced [36,86,88]. Moreover, IFN-α and IL1-β inhibit HSC proliferation in vitro whereas IFN-γ, TNF-α and TLR ligands directly accelerate HSC proliferation in vitro [36,37,78,80,86], indicating that distinct signals promote HSC proliferation via different mechanisms. For instance, it has been demonstrated that a transcriptional suppressor of type I IFN signaling, interferon regulatory factor-2 (IRF2), negatively regulates HSC proliferation and Irf2^–/–^ HSCs show enhanced cell cycling status [79].

Increased HSC proliferation under inflammation is rapid but transient [37,78,89]. For instance, HSCs quickly return to a quiescent state, and p27 and p57 return to steady-state levels during in vivo chronic exposure to IFN-α [78]. Therefore, HSCs maintain a largely quiescent state during chronic inflammatory stress induced by polyinosinic-polycytidylic acid (poly I:C) [78], mimicking type I IFN-mediated response, IL1-β and chronic inflammatory arthritis, regardless of cell cycle activation at early phase of treatment [28,37,78,89]. This quiescent state under chronic inflammation is due to the repression of cell cycle and protein synthesis genes, which are mediated by activation of the transcription factor PU.1 and direct PU.1 binding at repressed target genes. Consistently, PU.1-deficient HSCs displayed overexpressed cell cycle and protein synthesis genes [82]. Aged HSCs display cell cycle arrest [90,91], which is associated with replicative stress [90]. Further studies are needed to investigate to what extent the dormant status of aged HSCs is caused by chronic exposure to inflammation.

### 3.2. Chronic Inflammation Triggers Myeloid-Biased Differentiation and Impaired Self-Renewal

Inflammatory signals activate HSC and promote myelopoiesis [37,75,83,84,85,92]. This response is beneficial in combatting infection, but chronic exposure to inflammatory insults impairs HSC self-renewal and causes stem cell loss. Most inflammatory stimuli have been reported to affect HSC multi-lineage differentiation and long-term repopulation potential (Table 2). HSCs grown in liquid culture with IL-1β produced more mature myeloid cells [36]. Furthermore, mice that were chronically exposed to IL-1β displayed increased myeloid cells at the expense of lymphoid cells. HSCs isolated from these mice displayed myeloid-biased differentiation potential and significant reduced self-renewal [36]. Of note, the impairment of HSC recovered after treatment, which is probably due to the reestablished quiescence. TNF-α also promotes myeloid regeneration in vitro [37,93]. Although no major changes in lineage distributions were observed from TNF-α-treated HSCs, these HSCs had severely compromised reconstitution abilities, which, similar to effects of IL-1β, recovered upon extra resting periods [36,37]. This demonstrates a transient impairment of the engraftment potential of IL-1β and TNF-a-treated HSCs [36,37]. Consistently, acute lipopolysaccharide (LPS) also induced transient changes in hematopoiesis, affecting epigenetic modifications and HSC gene expression [94]. Mice transplanted with LPS pre-stimulated HSCs displayed high survival against secondary bacterial infection [94]. However, chronic LPS treatment attenuated HSCs’ self-renewal and competitive repopulation activity [86]. Thus, HSCs respond differently to acute and chronic inflammation, and only chronic and continuous inflammation mimics the aging-associated functional declines. 

### 3.3. Inflammation-Associated Signals Are Activated in Aged HSCs

To date, we have limited knowledge of the mechanisms by which inflammatory signals regulate HSC function. Under chronic LPS exposure, the functions of HSCs were impaired in a TLR4-TRIF-ROS-p38-pathway dependent manner [86]. C/EBPβ is required for LPS-induced memory, which improves myeloid differentiation and the resistance to secondary infection [94]. The loss of C/EBPβ attenuates an IL-1β-driven myeloid gene program and expands hematopoietic stem and progenitor cells (HSPCs) [92]. It also has been shown that the induction of myeloid differentiation by IL-1β and TNF-α is likely due to the activation of PU.1 [36,82,93] and mice lacking the PU.1 upstream regulatory severely attenuated myeloid differentiation. The overexpression of PU.1 has been shown to accelerate the myeloid output of HSCs in vitro [36]. In addition, the TNF-α-dependent activation of PU.1 is directly regulated via NF-κB-dependent signaling [37,93]. Actually, the transient impairment of HSCs induced by TNF-α correlates with both cell cycle activation and the status of the NF-κB pathway [37], suggesting that this pathway is of vital importance for inflammatory hematopoiesis. Interestingly, NF-κB was shown to become activated in aged HSCs, documented by elevated phosphorylation and translocation in the nucleus [15,42,95]. This suggests that an active inflammatory response exists in aged HSCs at steady state and raises the possibility that NF-κB signaling pathway is a potential target to achieve rejuvenation of aged HSC.

### 3.4. Aged BM Niche Is Inflamed 

Considering the fact that pro-inflammatory cytokines are elevated in aged BM, the aged niche is also an inflamed niche. Inflammation can remodel the BM niche as niche cells themselves express various inflammation-associated receptors and thus contribute to HSC aging indirectly.

Exposure to LPS or poly I:C has shown to trigger bone marrow angiogenesis with an increased number of sinusoids, an increase in integrin αVβ3 expression and activation on ECs and vascular leakiness [96,97]. BM ECs, expressing high level of *Tlr4* and myeloid primary response gene 88 (*Myd88*), are the primary source of granulocyte colony-stimulating factor (G-CSF), the key granulopoietic cytokine, after LPS challenge or *Escherichia coli* infection. Therefore, ECs are essential cells for emergency granulopoiesis under systemic bacterial infection [98]. Consistent with this, young HSPCs cocultured on aged ECs acquired a myeloid bias with a decrease in B and T cell frequencies, and an in vivo infusion of aged endothelium into young recipients impaired HSC self-renewal and induced myeloid bias [65].

Aged BM stroma cells show increased expression of inflammatory chemokines (*Cxcl2* and *Cxcl5*) and several members of the complement cascade (*Cfd*, *Cfb*, *C4b*, and *C3*). Most importantly, *Il1b* and *Il6* expression levels are increased [72]. Likewise, aged BM macrophages also showed upregulation of *Il1b* [3]. These data are consistent with the accumulation of both cytokines in the aged BM [3,4], which supports the positive feedback that BM niche cells respond to inflammation to secrete more inflammatory cytokines, which in return enhance pro-inflammatory responses of niche cells (Figure 1). The expression of multiple pathogen sensors, *Tlr4* for example, and various effector molecules, including *Erk1*, *Elk1* and *Tbk1*, were increased in old plasma cells, indicating that old plasma cells were primed for TLR mediated inflammatory response [71]. Functionally, aged plasma cells stimulated myelopoiesis, and inhibited lymphopoiesis, when cultured with HSCs ex vivo and in vivo plasma cell depletion, reversed the age-associated enhancement of myelopoiesis [71].

### 3.5. Heterogeneous Response of HSCs to Inflammation 

Considering the fact that pro-inflammatory cytokines are elevated in aged BM. We described earlier how HSCs can directly respond to inflammatory stimuli as they express multiple receptors known to interact with inflammatory ligands. This raises the notion that heterogeneous responses of HSCs to inflammation may result if those receptors are heterogeneously expressed on HSCs. As discussed above, HSCs are phenotypically and functionally heterogeneous. Thus, there may be subsets of HSCs that show different responses to inflammatory insults. Single cell RNA sequencing has identified HSC subsets with distinct transcriptional responses to inflammatory signals [41]. For example, type I IFN, TNF, and IL-1β all expanded CD41^high^ and P-selectin^+^ (Selp^+^) HSCs. The high expression of CD41 coincides with stem-like megakaryocyte-committed progenitors within the HSC pool, and the expansion of this pool is associated with activated megakaryopoiesis [34,36]. Upon chronic LPS treatment, CD86^+^ HSCs, primed for lymphoid-biased differentiation, were reduced [48]. IL27Ra marks a population with impaired self-renewal and myeloid-skewing. This subset expanded when exposed to TNF-α. Of note, RNA-sequencing revealed that IL27Ra^+^ HSCs displayed inflammatory signatures in comparison with IL27Ra^-^ HSCs, indicating that there are HSC subsets that are primed for potential inflammatory stress during homeostasis [42].

IL-6R and TLR4 were shown to be more abundantly expressed by old myeloid-biased HSCs, indicating that aged HSCs sense inflammation signals differently compared to young HSCs. Indeed, aged HSCs have differential responses to inflammatory challenges compared to young HSCs [41]. Aged HSCs showed more skewed myeloid differentiation in vivo after 2 h culture with LPS, while young HSCs still maintained a balanced output [41]. Furthermore, the frequency of HSC subsets with heterogeneous transcriptional profiles to inflammatory signals also alters with age [41]. Specifically, inflammation-related genes were more enriched in the aged compared to the young IL27Ra^+^ HSCs, suggesting that IL27Ra^+^ HSCs are inflamed and the inflamed situation accumulates with age [42]. Collectively, this suggests that responses of HSC to inflammation changes with age as well. The changes in inflammatory responses with aging are likely due to the changes of HSC composition, where HSC subpopulations primed for inflammatory insults expand. Indeed, CD41 [38], Selp [5,15,52] and IL27Ra [42] are up-regulated with aging in HSCs (Table 1), which is largely due to the expansion of specific populations.

Most importantly, HSC subsets with inflammatory signatures displayed dysfunctions that are very similar to those induced by age. Collectively, a large part of age-associated HSC changes is likely caused by a dominance of inflamed HSCs that are further expanded by elevated inflammatory signals in the aged BM. In other words, inflamed HSCs may display age-associated functional decline and those inflamed HSCs are functionally “older”. In the aged BM, we suggest that less inflamed HSC subsets exist, which are functionally “young-like” and have a high stem cell potential.

## 4. Conclusions and Future Perspectives

The intimate association between HSC aging and age-associated chronic inflammation in the BM niche is becoming increasingly clear. The intrinsic mechanisms have been shown to be major contributors to HSC aging. Genes have been identified that correlate with, or in fact drive, age-associated HSC dysfunction. However, it is also evident that stem cell aging is a systemic process, with both intrinsic and extrinsic factors involved. We speculate that the intrinsic (i.e., transplantable) changes of HSC during aging may arise from a perturbed microenvironment, and we suggest that (acute or chronic) inflammation may be an important contributor to such perturbation. It should be emphasized that the impairment of HSC induced by chronic IL-1β exposure recovered upon secondary transplantation [36], while aged HSCs consistently display reduced repopulation capacities when serially transplanted. This indicates that the consequences of chronic inflammation do not capture all aging phenotypes and are not the sole extrinsic factors that contribute to HSC aging.

In addition to normal aging, recent studies have focused on the interplay between chronic inflammation, clonal hematopoiesis and hematological malignancies. Nonmalignant clonal hematopoiesis, which is referred to as clonal hematopoiesis of indeterminate potential (CHIP), is mainly driven by DNMT3A and TET2 mutations [19]. It has been demonstrated that chronic inflammation may be a driving the selecting force of HSC clones as it expands specific HSPC subsets and thus may contribute to CHIP [99,100]. For instance, recent data suggests that chronic IFN-γ signaling drives clonal hematopoiesis induced by DNMT3A loss of function [99]. In the development of several hematological diseases, including myeloproliferative neoplasms, myelodysplastic syndrome (MDS), acute myelogenous leukemia (AML) and chronic myelogenous leukemia (CML), various cytokines including IL-1, TNF, IL-6 and IFNs are involved [101,102]. It may be possible to intervene in inflammatory pathways and thus prevent or restore loss of function of aged hematopoiesis.

## Figures and Tables

**Figure 1 cells-10-01849-f001:**
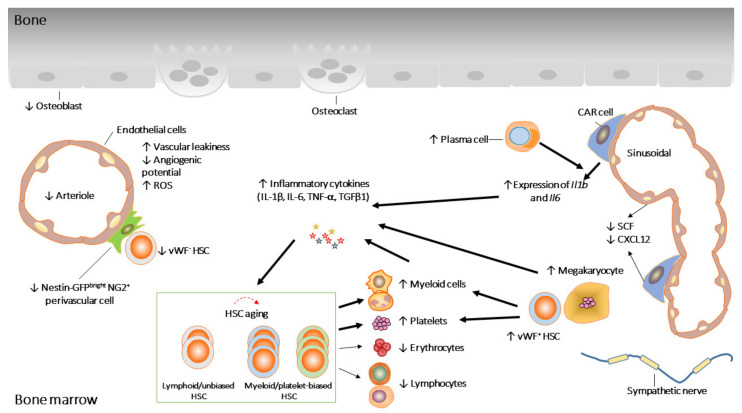
Age-associated changes in HSCs and their niche. Aged HSCs, with reduced repopulation potential, are dominated by myeloid/platelet-biased HSCs. HSCs’ age-associated defects are partially the consequences of age-associated changes in the niche. ↓ and ↑ represent increased and decreased activity, respectively.

**Table 1 cells-10-01849-t001:** Expression of HSC markers and their altered expression during aging or inflammation.

Markers	Self-Renewal	Bias	Change with Age	Change with Inflammation	Reference
CD150	High	Mye	Up	Up	[46,48]
vWF	High	Plt	Up	Up	[5,34,47]
CD86	High	Lym	Down	Down	[29,48]
Satb1	High	Lym	Down		[31,32,33]
CD9	High	Plt			[49,50]
CD38	High				[46]
Tie2	High	Bal			[51]
Jam2	High	Lym			[30]
CD41	Low	Mye/Plt	Up	Up	[34,36,37,38]
Selp	Low		Up	Up	[15,34,52]
Neo1	Low	Mye	Up	Down	[39,40]
CD61	Low	Mye	Up	Up	[34,41]
IL27Ra	Low	Mye	Up	Up	[42]
Alcam	Low	Mye	Up		[43,44]
c-Kit	Low	Plt			[45]

Mye/plt: myeloid- or platelet-biased; Lym: lymphoid-biased; Bal: balanced.

**Table 2 cells-10-01849-t002:** The role of pro-inflammatory cytokines in hematopoiesis and their altered expression during aging.

Stimuli	Source	Effects on HSC	Change with Age	Reference
IFN-α	Plasmacytoid dendritic cells, macrophages	Transient proliferationImpaired repopulation potentialExhaustion		[77,78,79]
IFN-γ	T cells, Th1 cells, macrophages	ProliferationImpaired repopulation potentialExhaustion		[80,81]
IL1-β	Monocytes, macrophages, ECs,	Myeloid differentiationImpaired repopulation potential	Up	[3,4,36,42,82]
TNF-α	Macrophages, T cells, NK cells	Myeloid differentiation	Up	[37,42]
IL-6	MSCs, macrophages	Myeloid differentiation	Up	[4,83]
GM-CSF	MSCs, ECs, macrophages, T cells	Myeloid differentiation		[75]
G-CSF	MSCs, ECs	Myeloid differentiation		[84,85]
TGFβ1	MSCs, Mks	QuiescenceExpansion of myeloid-biased HSC	Up	[2,69]
LPS	Gram-negative bacterial infections	ProliferationImpaired repopulation potential		[48,86]

ECs: endothelial cells, MSCs: mesenchymal stromal cells, Mks: megakaryocytes.

## Data Availability

Not applicable.

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
