# Peer review of "Inflammation and Aging of Hematopoietic Stem Cells in Their Niche"

_cells, 2021, doi:10.3390/cells10081849_

Round 1

Reviewer 1 Report

Daozheng and Gerald reviewed about latest knowledge for Hematopoietic Stem Cell and their Bone Marrow environment on Aging especially focused on Inflamation.

Great work to encourage reader to understand recent work around the area, though we have some comments to improve. 

FIGURE1, At HSC aging panel, that's difficult to catch up what’s explaining. Lymphoid/unbiased HSC self-renew more of decreased? What’s the difference between blue cells and green cell in Myeloid/platelet-biased HSC? Revise to understand easily.

LINE92, The section’s number should be 2.

LINE108, Author wrote “aged BM niche contributes to reduced self-renewal and skewed myeloid differentiation of aged HSCs”, but results showed both intrinsic and extrinsic of HSC affteced to aged phenotypes for HSCs.

LINE167, “Upon aging, both cell types(BM macrophages and plasma) displayed inflammation-associated changes” and what’s the result in HSC should be written.

LNE169 “as they age” should be “as their age”.

LINE191,  INFs should be IFNs.

Table2, Difficult to recognize which Stimuli for Effects on HSC, make a space between Stimulies.

LINE236, Not “Inflammation-associated mechanisms”, “Inflammation-associated signals” should be better to understand.

LINE316, Authers focused on Inflamation at Aging and they speculated repression of Inflamation would rejuvenate Aged HSCs. However, Aging should be contained more factors, DNA damage, low metabolism and changed epigenetic modifications e.g.. Consider that and rephrase would be better.

Reviewer 2 Report

SUMMARY

This review article focuses on the impact of aging on hematopoietic stem cells (HSCs) and also touches briefly on similarities between chronic inflammation and the aging phenotype. Overall, the authors have done a good job at stating the most recent facts on the topic. However, moving beyond factual statements to draw new connections (or highlight the lack thereof) for the reader will greatly improve the manuscript. Adding speculations throughout the manuscript regarding what the field is lacking in terms of knowledge (such as more comprehensive formal demonstrations of links between inflammation and aging), what needs to addressed and how it can be addressed will also add significant value of the manuscript to the field. For instance, further highlighting stromal inflammation (including bioRxiv paper from Passegue group, which implicates IL-1) and remodeling, and discussing potential impacts of cellular senescence and the SASP (a lot of interesting work from Judy Campisi and Daohong Zhou in this area that touch upon IL-1, as well as others) on HSC function and inflammatory metabolic/autophagy changes occurring with age that may impact hematopoiesis could add excitement and interest to the work by suggesting new lines of inquiry. Moreover, adding concluding summary statements to each topic is highly recommended to help with the flow of the manuscript as well as connect all the literature together.

TOPIC 1.1

  1. In general, the authors are missing the 3rd function of HSCs, which is the ability to engraft – it would be helpful to add details on this aspect and how it is impacted with aging, as stem cell transplantation is a well know therapeutic owing to the engraftment potential of HSCs.
  2. Line 20: The authors should cite more papers for the myeloid bias of HSCs with aging. It has been known since at least past 10 years, and 2019 (the papers cited) is not the first time this came to light. The same goes for lymphopoiesis in aging. (Christos Gekas et al, Blood 2013; Derrick J Rossi et al, PNAS 2005)
  3. Line 32: there is evidence that HSCs can also undergo asymmetric cell division. Kindly rectify the statement and add appropriate citations. (Brain Zimdahl et al, Nature Genetics 2014; Dirk Loeffler, Nature 2019)
  4. Line 33-34: is a repeat of what is already mentioned in paragraph 1. Since paragraph 2 mostly talks about self-renewal, these lines can be omitted and focus on one functional aspect of HSCs and it impact with age.
  5. Line 36-37: the authors are missing the fact that increased HSC pool size with age is also considered as an effect of increased HSC proliferation

TOPIC 1.2

  1. Line 59-60: more recent publications have shown a much stringent gating for HSCs. Those papers should also be cited here (Eric M. Pietras eta al, Cell Stem Cell 2016; Jennifer L Rabe et al, Experimental hematology, 2020)
  2. Line 63-64: it would be beneficial for the readers if the authors could give more information on the markers that are heterogeneously expressed on HSCs. Just stating the markers will add to the value of the sentence. OR combine and re-write lines 63-66 – they are giving the readers the same information
  3. Table 1: The authors should include CD9 in the table (Ayako Nakamura-Ishizu et al, Cell Reports 2018)

TOPIC 2

  1. This has been mis-numbered as 1 instead of 2
  2. In general, since studies are still underway as to how aged niche impacts HSCs, understandably the literature is still unavailable to include in the review. However, the authors viewpoints on the topic would be highly appreciated and add to the impact of the review article. For instance, in lines 139-140, the authors mention various characteristics of aged niche including increased ROS. There is literature of how ROS impacts HSCs, and hence speculations and hypothesis can easily be inferred. Similar inferences can be made at the end of all paragraphs in this topic.
  3. Line 125-126: It would be beneficial for the readers if the authors elaborate on where the aged HSCs are located if not by the arterioles.
  4. Line 134: the authors should mention the full name for EMCN, as this is the 1st time the abbreviation is used
  5. Line 137: the word ‘high’ and symbol ‘-‘ needs to go in superscript for CD45-Ter119CD31high

TOPIC 3

  1. Line 175: The authors should mention the chronic inflammatory stressor and whether the HSC phenotype is consistent for all chronic inflammation stressors or just specific to some?
  2. Line 176-179: What about the inflamed niche, does that also replicate the aged niche just like the HSCs?
  3. Line 197: IL-1 actually seems to inhibit HSC proliferation in vitro, at least the first round of cell division, based on the Pietras group studies.
  4. Line 213: The cell cycle status of aged HSC is thought to reflect G1/S arrest (Flach et al, 2014) related to replicative stress. It may not actually reflect inflammation-enforced quiescence.
  5. Line 216-218: The authors should add references for these statements.
  6. Line 239-241: The authors should also cite and mention the recent paper on impact of CEBPA-KO in HSCs in the context of inflammation (Kelly C Higa et al, Journal of experimental medicine 2021)
  7. Line 248: It seems notable that the extent to which ‘normal’ inflammatory pathways are active in aged HSC has not been systematically determined. For instance, activation of PU.1, NF-kB, aging-related signals such as Wnt5a, etc. may not be normal in aged HSC. For instance, if HSC in aged animals and humans are expanding, perhaps this argues against the activity of factors such as PU.1.
  8. The authors should consider merging and condensing section 3.5 with sections 3.1 and 3.2 as most of the section 3.5 is a repeat of what is already mentioned and the couple of new things mentioned can easily fit into sections 3.1 and 3.2
  9. It does not appear clear as to what extent aging is a reflection of responses to ongoing inflammation versus the cumulative impact of replicative history and associated changes in mitochondrial inheritance/activity (work from Kateri Moore, Isabel Beerman, Mick Milsom and Marie Philippi could be cited) are driving aging phenotypes in HSC. For instance, HSC expansion is not necessarily a feature of chronic inflammation (the expanding cells do not appear to be HSC based on work from Pietras and Essers), and certainly not to the extent observed in aging. The authors might consider in a critical light to what extent inflammation resembles and/or contributes to such key attributes of aged HSC. Drawing thoughtful contrasts in this setting could be helpful to the reader.
  10. Myeloid vs. Mk bias are important distinctions, as aged HSC appear to be profoundly Mk biased (Nerlov’s and Jacobsen’s papers). The argument appears to be that lymphoid activity is suppressed whereas Mk activity is amplified in aging – this may not be entirely congruent with the authors’ statements in 3.3, such as in lines 316-319.

CONCLUSION

The authors bring up CHIP mutations which has literature evidence for relevance to aging and inflammation. However, the connection is lacking in the concluding paragraph. It is recommended that the authors provide a brief background to CHIP mutations and inflammation/aging instead of jumping right into the topic. OR, the authors could also remove the CHIP mutations from the conclusion and focus on what is still unknown for the field of aging/inflammation in context of HSCs and how that can be clinically translated.

Reviewer 3 Report

This is an excellent review setting the niche-based chronic inflammation as a major causal factor for the aging of hematopoietic stem cells. I have only one question. Since most of the reported experiments and analyses are made in the mouse, the conclusions are valid for this species, but it is unclear how they can be extrapolated to humans. Can the authors comment on this issue.

Author Response

This is an excellent review setting the niche-based chronic inflammation as a major causal factor for the aging of hematopoietic stem cells. I have only one question. Since most of the reported experiments and analyses are made in the mouse, the conclusions are valid for this species, but it is unclear how they can be extrapolated to humans. Can the authors comment on this issue.

Thank you very much for this assessment. We believe what is most relevant to humans is the connection between chronic inflammation and clonal hematopoiesis. In the revised manuscript, we have added more explicit thoughts in this matter in the conclusion section. In general, published data indicates that chronic inflammation can expand HSC clones with DNMT3A and TET2 mutations. This could explain clonal hematopoiesis frequently happens in the elderly.